# SEMANTIC DIFFERENTIATION FOR TACKLING CHALLENGES IN WATERMARKING LOW-ENTROPY CONSTRAINED GENERATION OUTPUTS

## ABSTRACT

We posit and demonstrate that while the current approaches for language model (LM) watermarking are effective for open-ended generation, they are inadequate at watermarking LM outputs for constrained generation tasks like machine translation and abstractive summarization due to the lower entropy of the output space. We investigate the reasons for such shortcomings in a variety of prominent watermarking approaches, and propose an effective solution based on sequence-level watermarking with semantic differentiation to watermark LLM outputs for constrained generation tasks that balances the output quality, watermark detectability, and imperceptibility of the watermark. Specifically, we show that token-level watermarking algorithms that modify logits over vocabulary during autoregressive generation, fail because they under-utilize the sequence entropy of the LM available for watermarking constrained generation outputs. While sequence-level semantic watermarking algorithms are promising alternatives for exploiting the higher sequence entropy compared to the low-levels of token-wise entropy, we identify a different fundamental drawback termed *region collapse* in the operationalization of current approaches that causes poor watermarking performance. Current approaches pseudorandomly partition the sequence-level representation space into valid and invalid regions for watermarking, but their operationalization encourages most high-quality output embeddings to all collapse into a single region causing a trade-off in output quality and watermarking effectiveness. To mitigate this, we devise a scheme *SeqMark* to differentiate the high quality output subspace and partition it into valid and invalid regions for watermarking, ensuring the even spread of high quality outputs among all the regions for effective watermarking without compromising the output quality. SeqMark substantially improves watermark detection accuracy (up to 28% increase in $F_1$) while maintaining high generation quality in constrained generation settings.

## 1 INTRODUCTION

As progress in language modeling leads to increasingly human-like automatic text generation, demand for tracing the provenance and life-cycle of digital text on the internet has soared. Questions around plagiarism (Sullivan et al., 2023), copyright infringement (Zhong et al., 2023), veracity of text (Augenstein et al., 2024), misinformation (Gravel et al., 2023), multi-agentic communication, and many other legal and contractual frameworks require the ability to attribute text to its source. Language model watermarking – embedding a traceable digital marker in the language model's distribution over language – has emerged as a potential solution to address this need. Seminal work on watermarking Kirchenbauer et al. (2023) modifies the logits produced over the vocabulary at each generation step from a language model to prefer or disprefer certain tokens. Much of the follow-up work (Fernandez et al., 2023; Zhao et al., 2023; Takezawa et al., 2025) on this token-level watermarking framework has focused on improved robustness to edits, generation quality, detectability and other desiderata for watermarking. While effective for open-ended long-form generation, LLM watermarking has remained to be a challenging problem in low-entropy setups like factual question-answering and code generation. While some work (Lee et al., 2024; Lu et al., 2024) addresses this problem for code generation via selective token-entropy based watermarking, *we posit that in*

*general, all token-level watermarking schemes struggle in task settings with limited entropy*. For example, tasks like machine translation and summarization, although lower entropy than open-ended generation, still admit multiple feasible responses with sufficient *sequence level entropy* to embed a watermark, but token-level watermarking approaches fail to utilize this sequence-entropy effectively.

An appealing alternative paradigm of sequence-level watermarking was explored in recent work Hou et al. (2024a;b): instead of randomly partitioning the token space at each generation step, they focus on partitioning the manifold of entire sentences via hyperplanes into red/green regions, and then perform rejection sampling to watermark the generation by sampling from the green regions. While promising, *we identify a common issue of red/green region collapse* with these approaches that prevents low-entropy watermarking. The operationalization of these algorithms via locality sensitive hashing or k-means clustering leads to assignment of semantically similar sentences to the same red/green partition. Due to semantic diversity, this is not a problem for open-ended generation tasks, but for constrained generation tasks, most of the admissible outputs tend to be semantically close to one another and mostly collapse to a single region. This imposes a challenging tradeoff between output quality and watermark verifiability for watermarking in constrained generation tasks.

Therefore, we propose a novel sequence-level watermarking approach *SeqMark* that utilizes the limited sequence entropy effectively to watermark constrained generation tasks. This approach first identifies the semantic manifold of high quality admissible outputs for the task at hand, and then semantically differentiates between the elements of this manifold to propose pseudorandom red/green partitions in a manner such that the potential high-quality responses are evenly spread across the partitions, enabling successful watermarking in low-entropy settings. In this work, we empirically support our claims and observations on watermarking previously underexplored constrained generation tasks like machine translation and summarization. We demonstrate the issues with various token-level watermarking algorithms and sequence-level algorithms in the prior work and find that our proposed watermarking approach is more effective in constrained generation settings, while maintaining high watermarking capabilities in open-ended generation settings.

## 2    PRELIMINARIES

Watermarking, much like steganography, is inherently a problem of incorporating a digital marker into the (ideally noise-tolerant) carrier signal. Typically, two desiderata characterize watermarking: imperceptibility and robustness. Robustness refers to the difficulty of tampering with the watermark by altering the signal and imperceptibility refers to preserving the quality and perception of the signal. We are primarily interested in approaches that incorporate the digital marker into the language model's distribution and thus perform watermarking while generating text from the language model. We consider autoregressive language models parametrized by $\theta$ that estimate a probability distribution of the next token following a prefix $p_\theta(w_t \mid w_{<t})$, and use it to generate sequences token-by-token in a left-to-right manner.

**Token-level logit Watermarking**    Kirchenbauer et al. (2023) introduced a seminal token-level watermarking algorithm dubbed KGW that used the logits produced over vocabulary for each token during generation to embed the digital marker. Many following attempts (Liu et al., 2024; Fernandez et al., 2023) proposed variations to improve robustness and undetectability of the watermarking scheme. All of these approaches modify the token-level distribution for watermarking and generate accordingly. At an abstract level, during generation at each time step $t$, such approaches pseudorandomly partition the vocabulary $\mathcal{V}$ into a green list of size $\gamma|\mathcal{V}|$ and red list size $(1-\gamma)|\mathcal{V}|$ for a hyperparameter $\gamma \in (0, 1)$, using a hash from the previous token $w_{t-1}$ as random seed. The generation procedure is then modified to prefer the green-list partition over the red-list partition, thus embedding the watermark. Typically detection involves counting the number of green-list tokens in the query text and performing hypothesis testing to decide whether the presence of the green-list tokens is by chance or not.

**Sequence-level Watermarking**    To improve the robustness of the watermarking method to paraphrase attacks, a different line of approach dubbed as semantic watermarking (Hou et al., 2024a;b) focuses on using sequence-level distribution induced by the language model instead of token-level distributions to embed the watermark. In addition to robustness, this general approach is more well-suited for watermarking low-entropy constrained generation tasks. In general, the sentence-

embedding space (instead of the vocabulary) is pseudorandomly partitioned into green and red regions. Then a sentence is generated from the language modeling via rejection sampling until the resulting sentence embeddings falls in the green region or the budget for resampling is exhausted. For generating another sentence, another round of pseudorandom partitioning of the embeddings space using the hash based on the previously generated sentence is performed followed by rejection sampling for the next sentence. Prior work SemStamp (Hou et al., 2024a) uses locality-sensitive hashing (LSH) to pseudorandomly partition the embeddings space: LSH first samples $n$ random vectors from a normal Gaussian distribution to specify $n$ hyperplanes and thus $2^n$ regions. Given a hyperparameter ratio $\gamma$ and the hash of the previous sentence, the regions are partitioned into $\gamma 2^n$ green regions and $(1 - \gamma)2^n$ red regions. A sentence with embedding $v \in \mathbb{R}^d$ would receive an $n$-bit binary LSH signature $c$, where each bit specifies the location of $v$ with respect to each hyperplane – thus $c$ identifies the region assignment for $v$. In contrast, $k$-SemStamp (Hou et al., 2024b) uses k-means clustering for partitioning instead of LSH: it first estimates $K$ centroids for a general semantic manifold via pretraining, and then during the generation step it randomly assigns each of these centroids either a red/green tag. During sampling, each generated sentence is assigned red/green tag according to its closest centroid. For both SemStamp and k-SemStamp, detection is done in a similar manner as token-level watermarking detection – it hinges on the membership of the query's sentences in the green region.

## 3 ENTROPY CONSIDERATIONS FOR IMPERCEPTIBLE WATERMARKING CONSTRAINED GENERATION: TOKEN-LEVEL VS. SEQUENCE-LEVEL

While we consider both imperceptibility and robustness to be important criteria for watermarking, *imperceptibility* of the watermark is very important for most textual applications in which the flow, tone, and naturalness of the text determine the experience of the reader. Imperceptibility requires redundancy in the signal, which for our purposes indicates the need for the distribution over text induced by the language model to have high entropy. From this perspective, constrained generation tasks are more difficult to watermark because the entropy over the desired text distribution is much lower than the case of open-ended generation.

Consider constrained generation tasks like machine translation and summarization: given an input, there is a small set of acceptable outputs that convey similar meaning but differ greatly in terms of style, syntax, coverage etc. This set is much smaller than open-ended generation but still larger and more varied than some low-entropy tasks like factual question-answering or even code-generation. Therefore, we posit and empirically observe with our proposed approach (§ 7), that these seeming low-entropy constrained generation tasks have enough sequence-level entropy to enable imperceptible watermarking. Assuming max-length of sequences (denoted by r.v. $\mathbf{y}$) is $T$, we can compute this entropy over $T$ random variables (tokens) under our model (parametrized by $\theta$) via chain rule for entropy: $H_\theta(\mathbf{y}) = \sum_{i=1}^{T} H_\theta(\mathbf{y_i}|\mathbf{y_{<i}})$ where crucially the conditional entropy involves summation over all the possible assignments of the prefix context $\mathbf{y_{<i}}$ i.e. $H(s \mid t) = -\sum_{s,t} p(s,t) \log p(s \mid t)$. This entropy is typically intractable to compute and is estimated (Kuhn et al., 2023) via sampling. In a sharp contrast, token-level entropy that is often used to characterize uncertainty (Duan et al., 2023) of language models is tractable to compute because instead of summing over all the possible prefixes, it commits to a single sampled prefix and computes the entropy over the possible tokens at the next step: $H_\theta(\mathbf{y_t}|y_{<t}) = -\sum_{w \in \mathcal{Y}} p(w \mid y_{<t}) \log p(w \mid y_{<t})$. The sequence entropy that is typically computed using token-level entropy computation for a sequence $w = w_1, , w_T$ given by $H_\theta(w) = \sum_{i=1}^{T} H(w_i \mid w_{<i})$ is one-sequence approximation to the full intractable sequence-level entropy. Therefore, while tractable and unattractive, common approaches of using token-level entropy to characterize sequence entropy tend to underestimate the true entropy over the sequences under the language model.

This observation has crucial ramifications for watermarking algorithms in low-entropy settings. The prevalent entropy-sensitive watermarking algorithms in prior work (Lee et al., 2024; Lu et al., 2024) only consider token-level entropy while embedding watermark in tokens during generation. For example, during tokenwise generation, Lee et al. (2024) only chooses to mark the tokens that have high entropy under the language model. As we show in our experiments (§ 7), this approach does not perform well for constrained generation tasks because it underutilizes the sequence-level entropy for embeddings watermark. It is interesting to note that token-entropy based watermarking

approaches Lee et al. (2024) perform favorably for code generation but fail at other constrained generation tasks. This is possibly due to code generation tasks having low sequence-level entropy that is well approximated by individual token-level entropy. But for tasks like machine translation and summarization, the need for exploiting higher sequence entropy becomes apparent. In Figure 1, we compare the KGW Kirchenbauer et al. (2023) token-level watermarking algorithm (blue) against two sequence-level watermarking algorithms, SemStamp Hou et al. (2024a)(orange) and our SeqMark approach (green) on the machine translation and summarization tasks. By sweeping over hyperparameters for these algorithms, we obtain a pareto curve for each approach. It is clear that the the token-level watermarking approach is inferior to both sequence-level approaches – we see a severe drop-off in watermark detection accuracy as the output quality increases for both the tasks. As a token-level approach commits to a sequence during left-to-right generation, it ignores all other

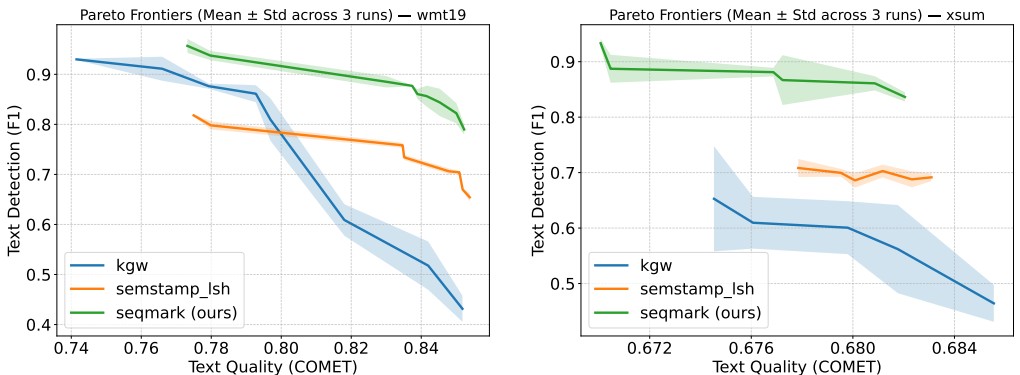

Figure 1: Pareto frontiers of token-level (Blue) watermarking, and two sequence-level watermarking approaches (Green and Orange) for machine translation (left) and summarization (right).

prefixes and potential paths that could lead to feasible outputs which drastically limits the entropy it can exploit for watermarking. Sequence-level semantic watermarking on the other hand considers sequence-level distribution via rejection sampling and is more amenable to exploit the sequence entropy afforded by constrained generation tasks.

## 4  REGION COLLAPSE: PITFALL OF STANDARD SEQUENCE-LEVEL WATERMARKING FOR CONSTRAINED GENERATION

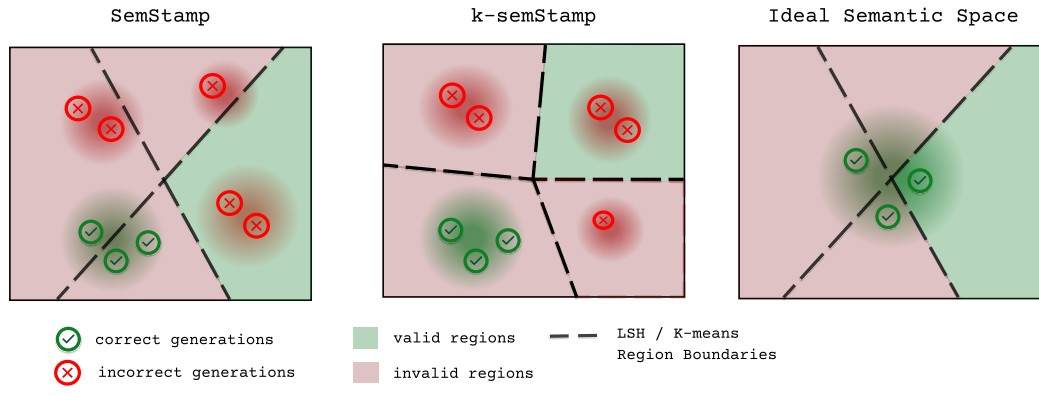

Figure 2: Semantic Space illustration for different sequence watermarking algorithms. Both Sem-Stamp **(Left)** and K-SemStamp **(Middle)** suffer from region collapse. **(Right)** Ideally, we would like to isolate the high-quality output manifold and partition it.

We observe in Figure 1 that sequence-level algorithm SemStamp behaves better than the token-level algorithm KGW, but still is inferior to our approach described later. We identify a crucial property of existing sequence-level watermarking algorithms like SemStamp that render them unsuitable for constrained generation watermarking. We call this property *region collapse* which refers to the effect that all viable outputs for a constrained generation task either randomly fall in the invalid red region or the valid green region. Consider the task of translation: If all the good translations of the prompt fall in the red region, then the sequence level watermarking algorithms will either generate a bad translation from the green region hurting the translation quality or exhaust it's resampling budget and output a translation from the red region hurting the watermarking effectiveness. Below we describe how region collapse is manifested in existing sequence-level watermarking algorithms: SemStamp (Hou et al., 2024a) and k-SemStamp (Hou et al., 2024b).

A well-trained language model induces a high probability on the correct responses for a constrained generation prompt and samples them more frequently. Also, because of semantic similarity of the high quality responses, the embeddings for these responses tend to have high similarity. An ideal sequence-level watermarking algorithms would produce red/green regions that separate these nearby responses to prevent region collapse as depicted in figure 2. However, *SemStamp*'s partitioning behaves in a manner *opposite* to this desired behavior as it employs locality sensitive hashing (LSH) to determine the partitions.

Under LSH (Charikar, 2002; Indyk & Motwani, 1998), given two vectors $x_i, x_j \in \mathbb{R}^d$ with angle $\theta_{ij} \in [0, \pi]$, the probability that they lie in the same region (LSH signature) is: $Pr[\text{region}(x_i) = \text{region}(x_j)] = \left(1 - \frac{\theta_{ij}}{\pi}\right)^d$. LSH determines partitioning hyperplanes such that similar points share the same partition, thereby *accelerating* region collapse. *k-SemStamp*, while shown to be

| Approach | Translation | Summary | Open-ended |
|----------|-------------|---------|------------|
| SemStamp | 0.190 / 0.973 | 0.392 / 0.916 | 0 714 / 0.587 |
| k-SemStamp | 0.246 / 0.973 | 0.106 / 0.942 | 0.872 / 0.309 |
| SeqMark | 0.812 / 0.007 | 0.916 / 0.027 | 0.957 / 0.006 |

Table 1: **Region entropy/average pairwise cosine similarity** across different tasks (columns) and watermarking approaches (rows).

more robust than SemStamp, unfortunately *exacerbates* the region collapse issue for constrained generation. As described in §2, this method explicitly focuses on assigning the same centroid/color to the semantically similar points with low embeddings distances, thus exacerbating region collapse.

As described later, we specifically address the issue of region collapse with our approach, SeqMark. In Table 1, we empirically demonstrate the issue of region collapse on the tasks of translation and summarization. We sample 100 high-quality (and high-probability) generations for each setting and report average pairwise cosine similarity and the estimated region entropy – a quantity that characterizes how evenly the high-quality points are spread across the partitions . While all three methods behave well for open-ended generation, for constrained generation tasks we observe low region entropy and high semantic similarity for SemStamp and k-SemStamp indicating region collapse, but high region entropy and low semantic similarity (explained below) for our SeqMark approach that explicitly fixes the issue of region collapse.

## 5 SEQMARK: AMELIORATING REGION COLLAPSE

As mentioned above, we devise a sequence-level watermarking approach that doesn't suffer from region collapse. We propose SeqMark, an approach that shares many similarities with other semantic watermarking approaches in that it pseudorandomly partitions the representation space into accept/reject regions and performs rejection sampling for watermarking, it differs significantly in how it operationalizes the partitioning procedure. As shown in figure 3, we first focus on isolating the subspace manifold (purple) containing high quality responses for the constrained generation prompt. We estimate this manifold by sampling highly-likely generations using low-temperature sampling assuming that high likelihood correlated well with response quality for well-trained LMs. Once isolated, we partition this manifold such that the high quality points are evenly distributed among the regions. To perform this partitioning, we still use LSH, but crucially we *transform the isolated manifold* such that its members (high-quality responses) are distant from one another, causing LSH to evenly spread them across the random partitions. Concretely, for an input prompt $p$ (and $t - 1$ generated sentences), we generate the response (t-th sentence $s_t$) as follows: We first sample $n$ high likelihood responses (with embeddings $c_i$) under low temperature to estimate high-quality

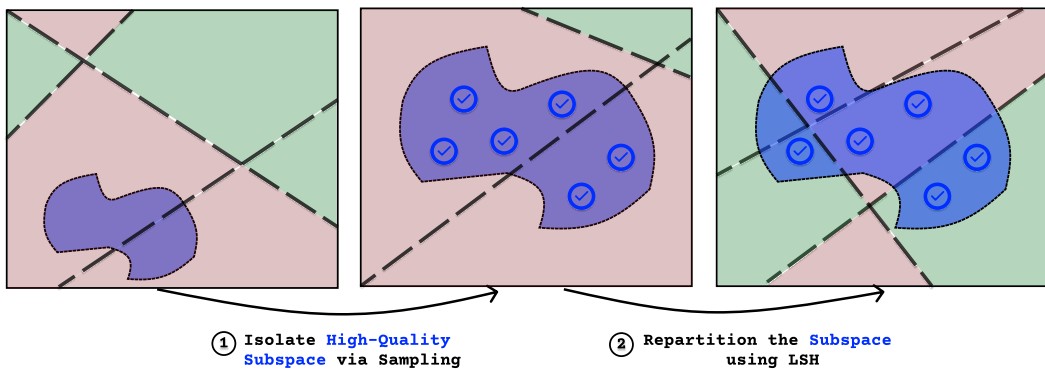

Figure 3: Depiction of SeqMark: Standard LSH in the first panel collapses the high quality output subspace(blue) to a small number of regions (red here). First, we identify the high-quality output subspace by sampling with low temperature; then we repartition by performing LSH on a *transformed* subspace designed to evenly spread it across multiple regions.

subspace $\mathcal{C} = \{c_1, c_2, ..., c_n\}$ Let $f(\cdot; \mathcal{C})$ be the function that transforms its members $c_i$ into a representation $u_i$: $u_i = f(c_i; \mathcal{C})$, that would be used to assign an LSH partition to $c_i$. To prevent region collapse, we aim to minimize the pairwise cosine similarity between the transformed members of $\mathcal{C}$ while preserving their relative proximity. This is difficult to estimate in general and could be approximated by learning such a function via a neural network. We opt for a much simpler choice of approximately but effectively estimating $f$: we simply subtract the sample mean is subtracted from each member of $\mathcal{C}$: $f(c_i; \mathcal{C}) = c_i - \frac{1}{n} \sum_{j=1}^{n} c_j$. We empirically observe that our mean-centering approach increases the distance between high-quality points in the desired manner (Table 1).

## 6 EXPERIMENTAL SETUP

**Tasks and Datasets**   We evaluate our approach on several constrained text generation settings: sentence translation with WMT19 German-English dataset (Bojar et al.), paragraph translation with WMT23 German-English (Kocmi et al., 2023), and abstractive summarization with XSum (Narayan et al., 2018). For unconstrained open-ended generation, we follow previous work and complete sentences from C4 RealNews dataset (Raffel et al., 2020). Table 8 presents the dataset details.

**Language Models and Embeddings**   For machine translation on WMT19, we use ALMA-7B (Xu et al., 2023) and LABSE (Feng et al., 2022) as the primary language model and sentence encoder, respectively. For paragraph translation, we use Gemma-2-4B-it (Gemma Team, 2024) due to its long context window. For other settings, we use Llama-2-7B-Chat (Touvron et al., 2023) language model and SBERT encoder (Reimers & Gurevych, 2019).

**Baselines**   We compare our approach against token-level watermarking approaches KGW Kirchenbauer et al. (2023), and SWEET Lee et al. (2024), and sequence-level watermarking approaches SemStamp Hou et al. (2024a), and k-SemStamp Hou et al. (2024b). As described above SWEET is a token-level approach specifically designed to address low-entropy watermarking. Given different sets of hyperparameters for each algorithm, we first sweep for the best hyperparameters over 100 samples to find the most promising configurations, then run them on the full evaluation sets (§A.1).

**Evaluation**   Two important dimensions for evaluating watermark algorithms are *text quality* and *watermark detectablity*. Text quality metrics often depend on the specific tasks and are thus described in the corresponding sections below. For evaluating watermark detection performance, we treat each query as a binary classification problem: either it was detected as a watermark or not. We then compute precision, recall, and $F_1$ based on detection prediction under various watermark algorithms. We report detection results for two types of negative examples: a) human (**h**) to evaluate differentiation between watermarked LM and human generated text, and b) non-watermarked LM

| | Translation (WMT19 De-En) | | | Summarization (XSum) | | |
|---|---|---|---|---|---|---|
| | COMET ↑ | P / R / F$_1$(h) | P / R / F$_1$(nw) | R-L ↑ / COMET ↑ | P / R / F$_1$(h) | P / R / F$_1$(nw) |
| *No Watermark* | *87.4* | - | - | *20.5 / 69.0* | - | - |
| KGW | 87.4 | 57.2 / 36.6 / 45.7 | 58.8 / 40.0 / 47.6 | 20.1 / 68.8 | 85.2 / 30.1 / 44.9 | 79.8 / 30.4 / 44.0 |
| SWEET | 87.2 | 57.5 / 30.0 / 39.5 | 58.1 / 30.1 / 39.6 | 18.5 / 68.7 | 52.4 / 49.9 / 51.1 | 53.0 / 40.0 / 45.6 |
| SemStamp | 87.4 | 59.5 / 73.7 / 65.9 | 58.7 / 74.0 / 65.5 | 20.0 / 68.7 | 70.6 / 52.8 / 60.4 | 67.8 / 59.0 / 63.1 |
| $k$-SemStamp | **87.5** | 63.3 / 33.0 / 43.4 | 62.2 / 33.0 / 43.1 | 20.7 / **68.9** | 54.2 / 22.0 / 31.3 | 61.6 / 22.0 / 32.4 |
| SeqMark | 87.1 | **76.9 / 77.3 / 77.1** | **75.5 / 83.0 / 79.0** | **21.6** / 68.5 | **81.3 / 100 / 89.7** | **85.3 / 85.3 / 85.3** |

Table 2: Watermarking results for sentence translation and summarization. Best results are **bold**. ↑ denotes the higher the better. For detection, **(h)** and **(nw)** denote the negative examples from human and non-watermarked LM, respectively. SeqMark substantially improves detection while maintaining competitive text quality.

(**nw**), to evalaute differentiation between watermarked generation and non-watermarked generation from the same language model.

# 7 MAIN RESULTS

**Sentence translation and summarization**   For these tasks, we primarily use COMET, a neural, semantic-based translation metric (Rei et al., 2020). Since COMET is not specifically trained for summarization, we also include ROUGE-L (Lin, 2004) for summarization results. Table 2 demonstrates that SeqMark substantially improves detection results while maintaining competitive text quality for constrained text generation. This result is congruent with Table 1, where the high cluster entropy score and low cosine similarity indicates that SeqMark correctly defines the high-quality output space and partitions it more uniformly for more effective watermarking. We observe that SWEET, a baseline explicitly designed to handle low token-level entropy performs very poorly on the constrained generation tasks highlighting the inability of token-level approaches to utilize the sequence entropy effectively. While SemStamp improves over token level approaches, we confirm our hypothesis that k-SemStamp exacerbates the region collapse issue by observing that it performs even worse than the token-level KGW approach. Finally, the trends are similar for both the types of negatives, human and non-watermarked LM, indicating that our watermarking approach successfully differentiates against texts from other sources.

**Paragraph translation**   We also evaluate our approach on paragraph translation, where the LM is tasked with translating multiple sentences. For this task, we concatenate 8-10 sentences from the same article in WMT23 dataset such that the input prompt fits within the context window of Gemma-2-4B-it. Since COMET is not specifically trained for paragraph evaluation, we also report BLEU (Papineni et al., 2002). Results in Ta-

| Approach | BLEU ↑/ COMET ↑ | P / R / F$_1$(h) |
|---|---|---|
| *No Watermark* | *39.2 / 87.1* | - |
| KGW | 40.7 / **87.7** | 70.4 / 38.0 / 49.4 |
| SemStamp | **42.8** / 87.5 | 55.6 / 100.0 / 71.4 |
| SeqMark | 39.8 / **87.7** | **90.9 / 100.0 / 95.2** |

Table 3: Paragraph translation watermarking.

ble 3 shows a similar trend as Table 2: SeqMark has significantly higher detection score than KGW and SemStamp, while maintaining similar text quality.

**Open-ended generation**   Similar to previous work, we evaluate watermarking algorithms with open-ended generations using C4 RealNews subset and report the results in Table 4. Expectedly, all watermarkign algorithms perform well in this high-entropy settings. Importantly, Seq-Mark performs on par with other watermark baselines without causing significant degradation in either text quality or text detection.

| Approach | PPL ↓ | P / R / F$_1$(h) | P / R / F$_1$(nw) |
|---|---|---|---|
| *No Watermark* | *3.4* | - | - |
| KGW | 3.6 | **100** / 92.0 / 95.8 | 94.1 / **95.0** / 94.5 |
| SemStamp | 3.6 | 98.9 / 94.9 / **96.9** | 98.9 / 93.9 / **96.3** |
| $k$-SemStamp | 3.6 | 96.4 / **95.0** / 95.7 | 97.6 / 93.9 / 95.7 |
| SeqMark | **3.4** | **98.9** / 94.0 / 96.4 | 98.9 / 92.0 / 95.3 |

Table 4: Open-ended generation with C4 RealNews

| Approach | COMET ↑ | Human | ALMA-7B | Gemma-2-4B-it | Llama-2-7B-Chat |
|---|---|---|---|---|---|
| KGW | **85.5** | 67.2 / 37.0 / 47.8 | 58.8 / 40.0 / 47.6 | 68.9 / 31.0 / 42.8 | 56.5 / 35.0 / 43.2 |
| SemStamp | 85.2 | 58.7 / 74.0 / 65.5 | 58.7 / 74.0 / 65.5 | 60.7 / 74.0 / 66.7 | 62.2 / 74.0 / 67.6 |
| SeqMark | 85.0 | **75.9 / 85.0 / 80.2** | **75.5 / 83.0 / 79.0** | **74.8 / 83.0 / 78.7** | **80.6 / 83.0 / 81.8** |

Table 5: Watermarking results on WMT19 translation tasks testing detection against negative examples from various sources: human and other non-watermarked LLMs.

**Detecting Watermarked LLM Against Other Language Models**  In addition to testing the ability of watermarking schemes to differentiate between watermarked texts and texts generated either by humans or non-watermarked LM, we also report results on the ability to differentiate between watermarked texts and texts generated from other (non-watermarked) language models. Using 100 samples from WMT-19 German-English as test bed, we compare different watermarking methods on ALMA-7B with unwatermarked completions from ALMA-7B, Gemma-2-4B instruct, and Llama-2-7B-Chat. Results from Table 5 indicate that SeqMark differentiates the watermarked text of a particular LM from other text sequences, be it human-generated or other LM-generated equally well.

# 8 FURTHER ANALYSIS AND DISCUSSION

## 8.1 EFFECT OF DIFFERENT TRANSFORMATIONS $f$ IN SEQMARK

The goal of the transformation $f$ in SeqMark is to reduce cosine similarity between high-quality generations. We propose a transformation that modifies sentence embedding by subtracting the sample mean i.e. $f(c_i; C) = c_i - z$, where $z = 1/n \sum_{c \in C} c$. In Table 6, we investigate other transformations obtained by subtracting vectors $z$ other than mean from the embeddings: random embedding, the sample closest to the mean, a single sample embedding $c$, the source sentence embedding, and the ground truth target translation embedding. We observe that all the transformations performed better than subtracting a random embedding in-

| Approach | COMET ↑ | P / R / F$_1$(h) |
|---|---|---|
| KGW | **85.5** | 67.2 / 37.0 / 47.8 |
| SemStamp | 85.2 | 58.7 / 74.0 / 65.5 |
| Sample mean | 85.0 | **75.9** / 85.0 / 80.2 |
| Random embedding | 85.4 | 51.3 / 19.0 / 27.7 |
| Sample closest-to-mean | 85.4 | 75.0 / **87.0 / 80.6** |
| Single sample | 85.3 | 71.1 / 59.0 / 64.5 |
| Source embedding | 84.9 | 67.9 / 53.0 / 59.6 |
| Target embedding | 85.4 | 64.8 / 59.0 / 61.8 |

Table 6: Results with different transformations (rows) in § 5, on WMT19.

dicating the need for preservation of relative relationships among the samples. Importantly, aggregated representations such as sample mean and point closest-to-the-mean perform the best, while single point embedding alternatives did not outperform SemStamp baseline.

## 8.2 FAST-SEQMARK

During generation, SeqMark needs to compute the sample high-quality outputs to estimate the sample mean for the transformation Consequently during detection, SeqMark computes the mean embedding via sampling requiring computation and access to the language model for detection. To avoid this, we propose a lightweight solution where a neural network is trained to map the input prompt to the sample mean embedding

| Approach | COMET ↑ | P / R / F$_1$(h) |
|---|---|---|
| KGW | **85.5** | 67.2 / 37.0 / 47.8 |
| SemStamp | 85.2 | 58.7 / 74.0 / 65.5 |
| SeqMark | 85.0 | **75.9 / 85.0 / 80.2** |
| Fast-SeqMark | 84.4 | 68.1 / 75.0 / 71.4 |

Table 7: Fast-SeqMark results on WMT19.

$z$ to subtract. We experiment with this Fast-SeqMark solution on the WMT19 German-English dataset. For training data generation, we use ALMA-7B to generate 50 samples each for 100,000 input prompts with a fixed temperature of 1.2. We then finetune the LABSE encoder on this dataset for mean prediction. Table 7 reports the result: compared to using the actual mean embeddings with SeqMark, Fast-SeqMark results in a drop in the detection score. Nonetheless, Fast-SeqMark still outperforms KGW and Semstamp.

### 8.3 PRACTICALITY OF SEQMARK

As discussed above, watermark detection for a query text requires access to the language model for sample generation to compute mean embeddings. While Fast-SeqMark is a viable alternative that alleviates this requirement, we posit that requiring such access for detection while not ideal, is not unreasonable. For example, prior work (Lee et al., 2024; Lu et al., 2024) also requires access to the LM during detection for computing token-level entropy. This access is readily available for open-weights models and could also be requested for realistic auditing purposes.

### 8.4 IMPERCEPTIBILITY WHEN WATERMARKING TRANSLATION OUTPUTS

Takezawa et al. (2025) recently showed that watermarking with minimal intervention is possible for machine translation tasks via their token-level NS-Watermarking method. When compared to our approach, we observed that this method achieves near perfect watermark detection performance but suffers in terms of quality measured by COMET with $\sim 6\%$ absolute decrease (see Table 2). Additionally, they acknowledge that by design thier approach is not imperceptible – it is easy to tell if the output has been watermarked. Looking at Table 11, it is easy to notice the minimal yet awkward token choices for watermarked translation. Concerningly, this imperceptibility makes this approach susceptible to simple post-editing attacks. On the other hand, our approach results in imperceptible watermarked generation making it robust to simple attacks.

## 9 RELATED WORK

**Language Model Watermarking**   Since its introduction, LM Watermarking has become an important technique to combat LLM-related security concerns (Suvra et al., 2023; Srinivasan, 2024). Most algorithms propose to add imperceptible statistical signals to different stages of text generation, including logit generation (Kirchenbauer et al., 2023; Zhao et al., 2023; Hu et al., 2023; Liu et al., 2024), token sampling (Christ et al., 2024; Kuditipudi et al., 2023), or even embed into the model weights during training (Sun et al., 2023; Gu et al., 2023). Orthogonally, several work propose sequence-level watermarking Hou et al. (2024a;b). The majority of work in LM watermarking concerns with open-ended text generation which affords high-entropy thus enabling highly effective watermarking (Ajith et al., 2024).

**Watermarking Low-Entropy Sequences**   Watermarking for constrained text generation tasks such as machine translation and summarization remains underexplored, despite these use cases constituting a non-trivial proportion of LLM usage. Most previous work on watermarking low-entropy sequences focuses on code generation: SWEET (Lee et al., 2024) extends KGW by selecting and watermarking high-entropy tokens, EWD (Lu et al., 2024) modifies the detection algorithm to include token entropy as weights in the final score, and Gu et al. (2025) effectively reduces watermarking efficiency without major loss in performance. For translation, NS-Watermark (Takezawa et al., 2025) extends KGW by observing the minimal number of watermarked tokens needed. In addition to superior empirical performance of our approach, our work also differs from the above works by focusing more on sequence-level watermarking paradigm instead of token-level watermarking.

## 10 CONCLUSION

In this work, we demonstrate the inadequacy of current watermarking algorithms for constrained generation tasks. We posit that all token-level algorithms perform poorly because they fail to utilize the semantic entropy induced by the LMs on these tasks. While sequence level watermarking algorithms are a better paradigm, we identify a different issue of region collapse in the operationalization of exiting semantic watermarking algorithms resulting in poor performance at watermarking constrained generation tasks. To overcome these limitations, we propose SeqMark, a sequence-level watermarking algorithm that ameliorates the region collapse issue by carefully partitioning the space for watermarking so as to ensure even spread of high-qualioty outputs among the partitions. We observe that our proposed scheme results in imperceptible watermarks that are reliably verifiable (detectable) while generating high-quality outputs for constrained generation tasks.

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

# A APPENDIX

## A.1 WATERMARK HYPERPARAMETER SWEEP FOR TRANSLATION AND SUMMARIZATION

Given different sets of hyperparameters for each watermark methods, to ensure apple-to-apple comparisons we sweep over a range of values of each method's relevant hyperparameters (Table 10). For each task, we sample 100 evaluation samples and run these watermark hyperparameter configs on these subset to obtain the Pareto curves in Figure 1. The best configs are then used to run the full evaluation set reported in Tables 2 and 4.

## A.2 DISCUSSION ON PARAPHRASING ATTACKS

A potential limitation of our approach is that, unlike SemStamp / k-SemStamp, it might be susceptible to paraphrasing attacks. By design, `SeqMark` spreads similar generations more evenly in the semantic space; thus a strong paraphraser can potentially evade the watermark by transforming the generation into another semantic region. However, in our experiments we observe that the paraphrasing outputs often have low text-quality, indicating that current paraphrasing models are not strong enough to maintain text quality, especially for low-entropy tasks like translations. In addition, existing sentence-level algorithms e.g. SemStamp does not have strong watermarking performance on these tasks, even without paraphrasing attacks (Table 2).

| Dataset | Task | Language | #Data | LLM | Sentence Encoder | Text Quality Metric |
|---------|------|----------|-------|-----|------------------|---------------------|
| WMT19 De-En | Machine Translation | German-English | 3000 | ALMA-7B | LABSE | COMET |
| XSum | Abstractive Summarization | English | 1000 | Llama-2-7B | all-mpnet-base-v1 | ROUGE-L / COMET |
| C4 RealNews | Open-ended Generation | English | 100 | Llama-2-7B | all-mpnet-base-v1 | Perplexity |
| WMT23 De-En | Long-form MT | German-English | 100 | Gemma-2-4B | LABSE | BLEU / COMET |

Table 8: Datasets, tasks, models, and evaluation metrics used in our experiments.

| **Translation Prompt** |
|---|
| Translate the following text from German to English: 
 German: München 1856: Vier Karten, die Ihren Blick auf die Stadt verändern 
 English: |
| **Summarization Prompt** |
| For the following article, write a one-sentence summary: "'The ex-Reading defender denied fraudulent trading charges relating to the Sodje Sports Foundation - a charity to raise money for Nigerian sport. Mr Sodje, 37, is jointly charged with elder brothers Efe, 44, Bright, 50 and Stephen, 42. Appearing at the Old Bailey earlier, all four denied the offence. The charge relates to offences which allegedly took place between 2008 and 2014. Sam, from Kent, Efe and Bright, of Greater Manchester, and Stephen, from Bexley, are due to stand trial in July. They were all released on bail. Summary: |

Table 9: Examples of prompt templates used in this work.

| Hyperparmeter | Methods | Values |
|---|---|---|
| Text generation temperature $t$ | KGW, SemStamp, $k$-SemStamp, SeqMark | [0.7, 0.85, 1.0, 1.2, 1.5] |
| Green/red list and regions ratio $\gamma$ | KGW, SemStamp, $k$-SemStamp, SeqMark | [0.1, 0.25, 0.5, 0.75] |
| Logit bias $\delta$ | KGW | [0.1, 0.5, 1.0, 2.0, 4.0] |
| LSH dimension $n$ | SemStamp, $k$-SemStamp, SeqMark | [2, 3, 4, 5] |
| High-quality cluster samples $c$ | SeqMark | [50] |

Table 10: Relevant hyperparameter for watermark methods

---

**Source**: München 1856: Vier Karten, die Ihren Blick auf die Stadt verändern
**Target**: Munich 1856: Four maps that will change your view of the city
**NS-Watermark**: Munich 1856: Four Engravings that Change Your View of the City
**SeqMark** : Munich 1856: Four maps that change your view of the city

---

**Source**: Kleingärtner bewirtschaften den einstigen Grund von Bauern.
**Target**: Allotment holders cultivate the soil of former farmers.
**NS-Watermark**: Allotment gardeners cultivate the former from of farmers.
**SeqMark** : Allotment holders cultivate the former fields of farmers.

---

**Source**: Es nervt, wenn Landkarten nicht aktuell sind.
**Target**: It is annoying when geographical maps are not up-to-date.
**NS-Watermark**: It's annoying when maps being out of date.
**SeqMark** : It annoys me when maps are not up-to-date.

---

Table 11: Three qualitative examples from WMT19. We include the source sentence, target sentence, NS-Watermark and SeqMark translations. We use underline to denote the watermarked tokens. Interestingly, the watermarked tokens in NS-Watermark are often the odd ones out in the translation (highlighted in red). We further note that NS-Watermark only uses beam search for decoding, which is fundamentally different from other approaches (including ours) that utilize token sampling.