# OpenReview forum: "Semantic differentiation for tackling challenges in watermarking low-entropy constrained generation outputs"
_ICLR.cc/2026/Conference — ICLR 2026 Conference Withdrawn Submission_

### Official Review · Reviewer_jRKe · 2025-11-01

**Soundness:** 3
**Presentation:** 3
**Contribution:** 3
**Rating:** 4
**Confidence:** 4

**Summary:**

The paper argues that existing LM watermarking methods are ill-suited for constrained generation because token-wise entropy is low even when sequence-level entropy remains usable. It diagnoses a core failure mode of prior sequence-level methods: region collapse, where semantically similar high-quality outputs cluster into the same red/green partitions, forcing a quality–detectability trade-off. The authors introduce SeqMark, which (1) samples likely high-quality candidates to estimate the “good output” subspace, (2) mean-centers this set to increase pairwise distances, and (3) applies LSH-based partitioning so good outputs distribute more evenly across regions; generation then uses rejection sampling on valid (“green”) regions. Empirically, SeqMark improves detection F1 by up to ~28% while keeping translation/summarization quality roughly unchanged; a lighter Fast-SeqMark predicts the mean embedding to avoid LM access at detection, trading some accuracy.

**Strengths:**

1. The “region collapse” concept is well-motivated and contrasted with LSH/k-means behavior in SemStamp/k-SemStamp, including a helpful schematic and an explanation of why constrained tasks exacerbate it.

2. The mean-centering transform is elegant, cheap, and shown to spread high-quality candidates across partitions, increasing “region entropy” and reducing cosine similarity among candidates

3. Evaluations cover sentence translation, paragraph translation, summarization, and open-ended generation. Open-ended results show SeqMark matches prior work without quality loss, suggesting no regression in high-entropy regimes.

**Weaknesses:**

1. Results are mainly point metrics (P/R/F1). ROC/PR curves, confidence intervals, and explicit operating thresholds across tasks would give a clearer picture of false-positive trade-offs, especially for human vs. non-watermarked-LM negatives.

2. This paper focuses on the entropy and the semantic watermark. However, the discussion of related work is limited. More related works should be discussed.

3. How sensitive is SeqMark to the encoder family (domain/language mismatch), and to alternative embedding spaces? A cross-encoder robustness study is missing.

4. SeqMark’s detector samples to estimate the candidate mean, implying access to the (watermarked) LM at verification time; Fast-SeqMark mitigates this but loses F1. A fuller analysis of deployment models (closed-weight LMs, third-party auditors) and security implications (e.g., keying, secret seeds, and adversary knowledge) would strengthen the case.

**Questions:**

Please see above.

---

> ### Author Response · Authors · 2025-11-21
> **Author Response to Reviewer jRKe**
>
> We thank you for acknowledging our strengths in the region collapse problem formulation, the elegant SeqMark solution, and the effective evaluations. We address your’s concerns below:
>
> 1.
> We reported precision, recall, F1 due to the nature of the evaluated tasks (translation and summarization). Since we only generate a handful of (often single) sentences per example, the z-test employed in sequence-level watermarking SemStamp is severely underpowered. In fact, for sequence-level detection these tasks are effectively binary classification: the sentence is watermarked or not. To ensure fairness of comparison between the baselines and our approach, we selected the threshold that yields the highest true positive rate / false positive rate ratio and reported the corresponding classification metrics.
>
> For completeness, we include the AUROC/TP@1/TP@5 for the results in Tables 1, 2 below. We note that the AUROC of token-level algorithms (KGW, SWEET) is not exactly comparable to the AUROC of sequence-level algorithm (SemStamp, SeqMark) in this case. We also reported AUROC/TP@1/TP@5 for tasks of paragraph translation (Table 3) and open-ended generation (Table 4).
>
> Table 1: Sentence Translation
> | Method           | BLEU / COMET |     P / R / F1 (h) | AUROC / TPR@1 / TPR@5 |
> | ---------------- | -----------: | -----------------: | --------------------: |
> | **No Watermark** |  38.1 / 87.4 |                  – |                     – |
> | **KGW**          |  38.0 / 87.4 | 57.2 / 36.6 / 45.7 |     55.9 / 7.1 / 18.8 |
> | **SWEET**        |  37.0 / 87.2 | 57.5 / 30.0 / 39.5 |      56.7 / 1.0 / 6.1 |
> | **SemStamp**     |  37.8 / 87.4 | 59.5 / 73.7 / 65.9 |     61.9 / 0.0 / 44.4 |
> | **k-SemStamp**   |  39.1 / 87.5 | 63.3 / 33.0 / 43.4 |     56.8 / 0.0 / 33.0 |
> | **Ours**         |  37.2 / 87.1 | 76.9 / 77.3 / 77.1 |     77.0 / 0.0 / 77.3 |
>
> Table 2: Summarization
> | Method           | ROUGE-L / COMET |     P / R / F1 (h) | AUROC / TPR@1 / TPR@5 |
> | ---------------- | --------------: | -----------------: | --------------------: |
> | **No Watermark** |     20.5 / 69.0 |                  – |                     – |
> | **KGW**          |     20.1 / 68.8 | 85.2 / 30.1 / 44.9 |    75.2 / 26.4 / 45.7 |
> | **SWEET**        |     18.5 / 68.7 | 52.4 / 49.9 / 51.1 |     52.8 / 2.9 / 10.7 |
> | **SemStamp**     |     20.0 / 68.7 | 70.6 / 52.8 / 60.4 |     65.4 / 0.0 / 52.8 |
> | **k-SemStamp**   |     20.7 / 68.9 | 54.2 / 22.0 / 31.3 |     51.7 / 0.0 / 22.0 |
> | **Ours**         |     21.6 / 68.5 |  81.3 / 100 / 89.7 |      88.5 / 0.0 / 100 |
>
> Table 3 Paragraph Translation Watermarking
> | Method           | BLEU / COMET | P / R / F1 (h)      | AUROC / TPR@1 / TPR@5 |
> | ---------------- | ------------ | ------------------- | --------------------- |
> | **No Watermark** | 39.2 / 87.1  | –                   | 46.3 / 0.0 / 2.0      |
> | **KGW**          | 40.7 / 87.7  | 70.4 / 38.0 / 49.4  | 72.1 / 11.0 / 25.0    |
> | **SemStamp**     | 42.8 / 87.5  | 55.6 / 100.0 / 71.4 | 84.0 / 30.0 / 40.0    |
> | **Ours**         | 39.8 / 87.7  | 90.9 / 100.0 / 95.2 | 100 / 100 / 100       |
>
> Table 4: Open-ended generation with C4 RealNews
> | Method           | PPL | P / R / F1 (h)     | AUROC / TPR@1 / TPR@5 (h) | P / R / F1 (nw)    | AUROC / TPR@1 / TPR@5 (nw) |
> | ---------------- | --- | ------------------ | ------------------------- | ------------------ | -------------------------- |
> | **No Watermark** | 3.4 | –                  | –                         | –                  | –                          |
> | **KGW**          | 3.6 | 100 / 92.0 / 95.8  | 99.0 / 92.0 / 95.0        | 94.1 / 95.0 / 94.5 | 97.9 / 99.0 / 100          |
> | **SemStamp**     | 3.6 | 99.0 / 95.0 / 96.9 | 98.7 / 90.9 / 97.0        | 98.9 / 93.9 / 96.3 | 99.0 / 85.9 / 98.0         |
> | **k-SemStamp**   | 3.6 | 96.4 / 95.0 / 95.7 | 94.8 / 80.0 / 87.0        | 97.6 / 93.9 / 95.7 | 94.8 / 80.0 / 87.0         |
> | **Ours**         | 3.4 | 99.0 / 95.0 / 96.9 | 99.3 / 94.0 / 97.0        | 98.9 / 92.0 / 95.3 | 99.0 / 94.0 / 96.0         |

---

> > ### Author Response · Authors · 2025-11-21
> >
> > 2. We will extend the Related Work section in the next iteration by elaborate more previous relevant work (e.g., CZG algorithm, UNIGRAM-WATERMARK) as well as discussing uncertainty estimation and the relationship between semantic entropy and empirical token entropy (http://arxiv.org/abs/2302.09664).
> >
> > 3. Our choice of encoder was task-dependent: LABSE encoder for machine translation and SBERT for summarization. We also experimented with other encoders and did not see major variations in results, indicating the robustness of SeqMark (See below). In addition, https://arxiv.org/pdf/2505.12540 shows that embedding models tend to learn similar embedding spaces. Thus, we believe our results are unlikely to change.
> >
> > | Method / Encoder                         | ROUGE-L / COMET | P / R / F1 (h)     | AUROC / TPR@1 / TPR@5 |
> > | ---------------------------------------- | --------------- | ------------------ | --------------------- |
> > | **SemStamp (encoder=all-mpnet-base-v1)** | 21.1 / 68.7     | 70.6 / 52.8 / 60.4 | 66.5 / 0.0 / 58.0     |
> > | **SeqMark (encoder=all-mpnet-base-v1)**  | 20.6 / 68.5     | 76.1 / 100 / 86.4  | 84.5 / 0.0 / 99.0     |
> > | **SemStamp (encoder=cde-small-v2)**      | 22.0 / 68.3     | 80.0 / 48.8 / 60.6 | 65.3 / 0.0 / 48.8     |
> > | **SeqMark (encoder=cde-small-v2)**       | 21.2 / 68.3     | 79.5 / 100 / 88.6  | 87.0 / 0.0 / 100.0    |
> >
> >
> > 4. As pointed out, we attempted to mitigate access to the original LMs during verification time with Fast-SeqMark. While Fast-SeqMark loses F1 compared to SeqMark, it still outperforms previous baselines by a margin. Alternatively, close-weight LLM providers can opt to save the candidate means during generation and subsequently provide them for verification. We leave the elimination of white-box LM access during detection to future work.
> >
> > We will update the submission PDF to reflect the above changes.

---

### Official Review · Reviewer_zKCx · 2025-11-01

**Soundness:** 2
**Presentation:** 3
**Contribution:** 1
**Rating:** 2
**Confidence:** 4

**Summary:**

This paper tries to address the challenge of applying language model watermarks to low-entropy generation tasks.
The authors posit that previously token-level methods (e.g., SWEET) are inadequate because they only consider token-level entropy and fail to utilize the much richer *sequence-level entropy* . While other sequence-level semantic watermarks (e.g., SemStamp, k-SemStamp) are better suited, the paper also identifies "region collapse" in these methods: in constrained tasks, all viable, high-quality outputs are semantically similar. SemStamp utilize semantic partitioning methods like LSH or k-means are designed to group similar items together, which causes all high-quality outputs to "collapse" into a single valid (green) or invalid (red) region, forcing a severe trade-off between output quality and watermark detectability. To solve this, the paper proposes a variant of SemStamp, which transforms the embedding spacek, allowing LSH to partition the "high-quality output subspace".

**Strengths:**

- The identification of "region collapse" is a novel and insightful critique of existing semantic watermarking methods.The empirical evidence in Table 1 (showing low region entropy for baselines) strongly supports this claim.

- The presentation is good and clear.

**Weaknesses:**

1.  The core premise of SeqMark is to "isolate the high-quality output subspace". However, the paper implements this by "sampling *n* high likelihood responses... under low temperature". The authors incorrectly and fatally equate **high probability** with **high quality**. This assumption is flawed. The method only ensures that *high-probability* samples are spread across partitions, not necessarily *high-quality* ones. This fails to solve the paper's stated problem: ensuring *high-quality* options are available in the "green" region. Also, the method was only tested on "easy" tasks (translation and summarization) where, for modern LLMs, the high-probability set is largely the high-quality set. This assumption would almost certainly fail in more complex, low-entropy domains like code generation or mathematical reasoning. The paper's claims of generalizability are unsupported without experiments in these challenging tasks.

2.  The paper argues that semantic aggregation methods (like SemStamp) are flawed due to "region collapse". However, it ignores that the entire purpose of semantic aggregation is to achieve **robustness** to paraphrasing attacks. SeqMark is built on the opposite principle of "semantic differentiation", which, as the authors admit in Appendix A.2, is more likely to be vulnerable to paraphrasing. The paper presents no experiments evaluating watermark strength under any attack (e.g., paraphrasing, editing, etc.). This is a major weakness.

3. The paper evaluates detection using Precision, Recall, and F1. These metrics are highly dependent on the classification threshold. The standard metrics for evaluating watermark strength are AUC-ROC or TPR@FPR (e.g., TPR@FPR=1%), which should be used for all detection results.

**Questions:**

1.  The definition of "constrained generation" is vague. Is this simply a synonym for "low-entropy tasks"? A clearer definition would be helpful.
2. How is "region entropy" (Table 1) calculated? The paper provides no formula or explanation.
3. Section 3 argues that token-entropy-based methods (like SWEET) fail because they under-utilize sequence entropy. However, the key experiment in Figure 1 uses KGW (a non-entropy-aware method) as the token-level baseline. To be consistent, the comparison should have been against SWEET

---

> ### Author Response · Authors · 2025-11-21
> **Author Response to Reviewer zKCx**
>
> We thank you for acknowledging the strengths of our work, particularly the clear presentation and the novel formulation of the region collapse problem. We address your’s concerns below:
>
> Weaknesses:
> 1. >“The authors incorrectly and fatally equate high probability with high quality. This assumption is flawed. The method only ensures that high-probability samples are spread across partitions, not necessarily high-quality ones.”
>
> -> We acknowledge that it might be imprecise to use “high-quality” to refer to high-probability generations. However, we believe this is a reasonable estimation for constrained generation tasks given the large(~infinite) output sequence space, especially under well-trained language models. For example, the math and code-oriented tasks often involve sampling high-probability samples from the models at a low temperature/greedy decoding with an implicit assumption that high probability most likely implies high quality. We do agree that this might not always be the case but it is not an unreasonable assumption for well-trained models. It is commonly observed that low-probability sequences tend to mostly be low-quality under today’s LLMs. In our experiments on translation and summarization, high-quality usually means high-probability as also recognized by you.
>
> >“Also, the method was only tested on "easy" tasks (translation and summarization) where, for modern LLMs, the high-probability set is largely the high-quality set. This assumption would almost certainly fail in more complex, low-entropy domains like code generation or mathematical reasoning.”
>
> In addition, we believe that watermarking performance on code or math generation tasks should not be adversely affected by this assumption based on the reasons you mention. Although we don’t provide empirical results on these tasks, as mentioned above, well-trained models tend to assign high-probability/scores to high quality answers. For example, SWEET (a popular tokenwise entropy-based watermarking algorithm) was evaluated on code generation with a low sampling temperature of 0.2 – that in itself is biased to high probability sequences.
>
> Nonetheless, we realize using the phrase “high-quality” when it means “high-probability” would cause confusion. As such, we will revise the paper accordingly and use high-probability everywhere because it is technically more accurate.

---

> > ### Comment · Reviewer_zKCx · 2025-11-27
> >
> > > the math and code-oriented tasks often involve sampling high-probability samples from the models at a low temperature/greedy decoding with an implicit assumption that high probability most likely implies high quality.
> >
> > This conflicts with the commonly used RLVR algorithms , where a higher temperature decoding is commonly used for rollout to ensure adequate exploration and discover genuinely high-quality examples.
> >
> > This assumption needs direct empirical validation. Arguing against the implementation details of the SWEET evaluation does not resolve the problem. Please run an experiment on the MATH500 benchmark (or a similar high-stakes reasoning benchmark) to demonstrate the performance of your method.

---

> ### Author Response · Authors · 2025-11-21
>
> 2.
> >“The paper argues that semantic aggregation methods (like SemStamp) are flawed due to "region collapse". However, it ignores that the entire purpose of semantic aggregation is to achieve robustness to paraphrasing attacks.”
>
> -> While it is true that existing semantic aggregation algorithms like semstamp and k-semstamp focus on paraphrase attack robustness, the major motivation behind our “semantic approach” to watermarking is a more serious issue focusing on appropriately using the limited sequence level entropy available for watermarking low-entropy constrained generation tasks like MT and summarization. This is evident from our results, wherein all the baselines including the extant semantic aggregation methods fail to effectively watermark the outputs for these tasks. We believe that absolute robustness to paraphrase attacks is a secondary concern in this scenario. Moreover, as we see in our results below, it is difficult to mount paraphrase attacks that do not affect output quality in constrained generation tasks in the first place!
> | Method                                                 | AUROC / TPR@1 / TPR@5 | BLEU / COMET |
> | ------------------------------------------------------ | --------------------- | ------------ |
> | **KGW**                                                | 60.5 / 6.0 / 22.0     | 35.6 / 85.5  |
> | **KGW (paraphrasing attacked)**                                     | 58.3 / 5.0 / 13.0     | 23.9 / 79.3  |
> | **SemStamp**                                           | 57.5 / 0.0 / 43.0     | 32 / 85.2    |
> | **SemStamp (paraphrasing attacked)**                                  | 46.5 / 0.0 / 21.0     | 13.9 / 75.4  |
> | **SeqMark**                                            | 76.5 / 0.0 / 86.0     | 34 / 85.0    |
> | **SeqMark (paraphrasing attacked)** | 53.7 / 0.0 / 37.0     | 21.8 / 80.1  |
> Data: WMT19 (100 examples). PEGASUS Paraphraser, following SemStamp
>
> >“SeqMark is built on the opposite principle of "semantic differentiation", which, as the authors admit in Appendix A.2, is more likely to be vulnerable to paraphrasing.”
>
> -> As you noticed, we have discussed the issues around paraphrase attacks at length in the appendix (section A.2) of our paper: A potential limitation of our approach is that, unlike SemStamp / k-SemStamp, it might be susceptible to paraphrasing attacks. By design, SeqMark spreads similar generations more evenly in the semantic space; thus a strong paraphraser can potentially evade the watermark by transforming the generation into another semantic region. However, in our experiments we observe that the paraphrasing outputs often have low text-quality, indicating that current paraphrasing models are not strong enough to maintain text quality, especially for low-entropy tasks like translations. In addition, existing sentence-level algorithms e.g. SemStamp does not have strong watermarking performance on these tasks, even without paraphrasing attacks.

---

> > ### Author Response · Authors · 2025-11-21
> >
> > 3.
> > We reported precision, recall, F1 due to the nature of the evaluated tasks (translation and summarization). Since we only generate a handful of (often single) sentences per example, the z-test employed in sequence-level watermarking SemStamp is severely underpowered. In fact, for sequence-level detection these tasks are effectively binary classification: the sentence is watermarked or not. To ensure fairness of comparison between the baselines and our approach, we selected the threshold that yields the highest true positive rate / false positive rate ratio and reported the corresponding classification metrics.
> >
> > For completeness, we include the AUROC/TP@1/TP@5 for the results in Tables 1, 2 below. We note that the AUROC of token-level algorithms (KGW, SWEET) is not exactly comparable to the AUROC of sequence-level algorithm (SemStamp, SeqMark) in this case. We also reported AUROC/TP@1/TP@5 for tasks of paragraph translation (Table 3) and open-ended generation (Table 4).
> >
> > Table 1: Sentence Translation
> > | Method           | BLEU / COMET |     P / R / F1 (h) | AUROC / TPR@1 / TPR@5 |
> > | ---------------- | -----------: | -----------------: | --------------------: |
> > | **No Watermark** |  38.1 / 87.4 |                  – |                     – |
> > | **KGW**          |  38.0 / 87.4 | 57.2 / 36.6 / 45.7 |     55.9 / 7.1 / 18.8 |
> > | **SWEET**        |  37.0 / 87.2 | 57.5 / 30.0 / 39.5 |      56.7 / 1.0 / 6.1 |
> > | **SemStamp**     |  37.8 / 87.4 | 59.5 / 73.7 / 65.9 |     61.9 / 0.0 / 44.4 |
> > | **k-SemStamp**   |  39.1 / 87.5 | 63.3 / 33.0 / 43.4 |     56.8 / 0.0 / 33.0 |
> > | **Ours**         |  37.2 / 87.1 | 76.9 / 77.3 / 77.1 |     77.0 / 0.0 / 77.3 |
> >
> > Table 2: Summarization
> > | Method           | ROUGE-L / COMET |     P / R / F1 (h) | AUROC / TPR@1 / TPR@5 |
> > | ---------------- | --------------: | -----------------: | --------------------: |
> > | **No Watermark** |     20.5 / 69.0 |                  – |                     – |
> > | **KGW**          |     20.1 / 68.8 | 85.2 / 30.1 / 44.9 |    75.2 / 26.4 / 45.7 |
> > | **SWEET**        |     18.5 / 68.7 | 52.4 / 49.9 / 51.1 |     52.8 / 2.9 / 10.7 |
> > | **SemStamp**     |     20.0 / 68.7 | 70.6 / 52.8 / 60.4 |     65.4 / 0.0 / 52.8 |
> > | **k-SemStamp**   |     20.7 / 68.9 | 54.2 / 22.0 / 31.3 |     51.7 / 0.0 / 22.0 |
> > | **Ours**         |     21.6 / 68.5 |  81.3 / 100 / 89.7 |      88.5 / 0.0 / 100 |
> >
> > Table 3 Paragraph Translation Watermarking
> > | Method           | BLEU / COMET | P / R / F1 (h)      | AUROC / TPR@1 / TPR@5 |
> > | ---------------- | ------------ | ------------------- | --------------------- |
> > | **No Watermark** | 39.2 / 87.1  | –                   | 46.3 / 0.0 / 2.0      |
> > | **KGW**          | 40.7 / 87.7  | 70.4 / 38.0 / 49.4  | 72.1 / 11.0 / 25.0    |
> > | **SemStamp**     | 42.8 / 87.5  | 55.6 / 100.0 / 71.4 | 84.0 / 30.0 / 40.0    |
> > | **Ours**         | 39.8 / 87.7  | 90.9 / 100.0 / 95.2 | 100 / 100 / 100       |
> >
> > Table 4: Open-ended generation with C4 RealNews
> > | Method           | PPL | P / R / F1 (h)     | AUROC / TPR@1 / TPR@5 (h) | P / R / F1 (nw)    | AUROC / TPR@1 / TPR@5 (nw) |
> > | ---------------- | --- | ------------------ | ------------------------- | ------------------ | -------------------------- |
> > | **No Watermark** | 3.4 | –                  | –                         | –                  | –                          |
> > | **KGW**          | 3.6 | 100 / 92.0 / 95.8  | 99.0 / 92.0 / 95.0        | 94.1 / 95.0 / 94.5 | 97.9 / 99.0 / 100          |
> > | **SemStamp**     | 3.6 | 99.0 / 95.0 / 96.9 | 98.7 / 90.9 / 97.0        | 98.9 / 93.9 / 96.3 | 99.0 / 85.9 / 98.0         |
> > | **k-SemStamp**   | 3.6 | 96.4 / 95.0 / 95.7 | 94.8 / 80.0 / 87.0        | 97.6 / 93.9 / 95.7 | 94.8 / 80.0 / 87.0         |
> > | **Ours**         | 3.4 | 99.0 / 95.0 / 96.9 | 99.3 / 94.0 / 97.0        | 98.9 / 92.0 / 95.3 | 99.0 / 94.0 / 96.0         |

---

> > ### Comment · Reviewer_zKCx · 2025-11-27
> >
> > - I strongly suggest replacing the paraphrasing method (PAGUSUS, an old LM dating back to 2019) with a GPT-based adversary [1]
> >
> > - I note that the reported AUC-ROC performance for all implemented watermarking methods is surprisingly low (Commonly it should reach over 90). Please clarify the average length of the generated text sequences used in this evaluation.

---

> ### Author Response · Authors · 2025-11-21
>
> Questions:
>
> 1. Definition of “constrained generation”: By constrained (or conditional) generation tasks we refer to tasks that are language-oriented, subjective, and have lower entropy than open-ended generation. We agree it is indeed difficult to arrive at the precise definition of the term as controlled, conditional, and constrained generation tend to be used interchangeably in literature as discussed here: https://arxiv.org/abs/2206.05395
>
> 2. Region entropy computation: Let R be a semantic region that a vector falls into. The region entropy is thus $-\sum_r p(r) \log p(r)$. We estimate p(r) using Monte Carlo sampling with N sampled sentences: $p(r) = 1/N -\sum_i^N 1(x_i \in r)$. We will add this clarification to the paper.
>
> 3. We reported an additional experiment against SWEET in Figure 1 of this supplementary document: https://anonymous.4open.science/r/semantic_watermark-8BC4/SeqMark_Supplementary_Materials.pdf
>
> We will update the submission PDF to reflect the above changes.

---

> ### Author Response · Authors · 2025-12-03
>
> * "This conflicts with the commonly used RLVR algorithms , where a higher temperature decoding is commonly used for rollout to ensure adequate exploration and discover genuinely high-quality examples."
>
> -> Our argument is for well-trained models, and a pre-RLVR model is not suitably trained for math/code tasks. After training, most models tend to have sharp probability peaks on a small number of (often correct) answers.
>
> * "This assumption needs direct empirical validation. Arguing against the implementation details of the SWEET evaluation does not resolve the problem. Please run an experiment on the MATH500 benchmark (or a similar high-stakes reasoning benchmark) to demonstrate the performance of your method."
>
> -> As requested, we provide results on a subset of the MATH500 dataset using OpenR1-distill-7B. Note that the watermark performances are higher across the board since the generations include long reasoning traces, which increase the empirical entropy. Experiments without reasoning traces would require a more careful experimental setup, since without the traces, the task accuracy would drastically decrease.
>
> | Method           | Accuracy | AUROC / TPR@1 / TPR@5 |
> | ---------------- | ------------ | --------------------- |
> | **KGW**          | 66.7 |  97.9 / 91.7 / 97.2  |
> | **SemStamp**     | 66.7 | 100 / 88.0 / 100.0   |
> | **SeqMark (Ours)**         | 66.7 |  100 / 100 / 100  |
>
> * "I strongly suggest replacing the paraphrasing method (PAGUSUS, an old LM dating back to 2019) with a GPT-based adversary [1]"
>
> -> We will re-run the paraphrasing analysis with GPT-based paraphrasers for the next draft
>
> * "I note that the reported AUC-ROC performance for all implemented watermarking methods is surprisingly low (Commonly it should reach over 90). Please clarify the average length of the generated text sequences used in this evaluation."
>
> -> The average generation length for sentence translation and summarization is about ~20 tokens per generation. For open-ended and paragraph translation, which are more suitable for AUROC/TPR@1/TPR@5 metrics, the average generation length is ~200 tokens per generation.

---

### Official Review · Reviewer_4sAV · 2025-11-01

**Soundness:** 3
**Presentation:** 2
**Contribution:** 3
**Rating:** 4
**Confidence:** 3

**Summary:**

The paper proposes a semantic watermarking method. In the partition of the embedding space, the proposed method aims to make high-quality generations fall in different red/green regions, which avoids the region collapse phenomenon that is observed for previous semantic watermarking methods. By doing the transformation of the embedding of the high-quality responses, the method decreases their cosine similarities and increases the region entropy. In machine translation and summarization tasks, the proposed method demonstrated better detection power while preserving similar generation quality.

**Strengths:**

The paper addresses the hard and important task of watermarking for LLMs in tasks with low-entropy generations. The paper provides intuitive explanations of the failure mode of existing semantic watermarking methods. Empirical results demonstrate the effectiveness of the proposed method

**Weaknesses:**

1. The paper would benefit from providing an algorithm with detailed descriptions of the full procedure. The proposed method uses locality sensitive hashing (LSH) to partition the embedding space, which is the same as some previous work. But it will still be beneficial to introduce how the partition works. The reject sampling generation and detection for red/green list type methods are relatively standard, but it would still be good to provide the detailed procedure, especially considering the general audience for the conference.

2. Some more explanation of why transforming $c_i$ by subtracting the mean can prevent region collapse would be beneficial.

3. No source code provided

There are also some minor typos, e.g., in line 293, "we simply subtract the sample mean is subtracted from"

**Questions:**

Please refer to weakness

---

> ### Author Response · Authors · 2025-11-21
> **Author Response to Reviewer 4sAV**
>
> We thank you for acknowledging the strengths of our paper, specifically on the difficulty of the task, as well as our problem formulation and the effective proposed solution. We address your’s questions below:
>
> 1. Following the request, we provide the full detailed SeqMark generation and detection algorithms in section B of this link: https://anonymous.4open.science/r/semantic_watermark-8BC4/SeqMark_Supplementary_Materials.pdf
>
> 2. We empirically demonstrated that mean-subtraction prevented region collapse in Table 1. We also provide the following theoretical justification in section A of this link: https://anonymous.4open.science/r/semantic_watermark-8BC4/SeqMark_Supplementary_Materials.pdf
>
> 3. We also provide the anonymous source code: https://anonymous.4open.science/r/semantic_watermark-8BC4/
>
> We will update the submission PDF to reflect the above changes (algorithms, theoretical justifications, as well as fixing the mentioned typos and other writing issues).

---

> > ### Comment · Reviewer_4sAV · 2025-11-26
> > **Response to Rebuttal**
> >
> > I appreciate the authors for the additional details in the algorithm and the theoretical justification for the proposed approach. I increased my score.

---

### Official Review · Reviewer_YEPP · 2025-11-01

**Soundness:** 4
**Presentation:** 4
**Contribution:** 3
**Rating:** 4
**Confidence:** 4

**Summary:**

This paper proposes SeqMark, a sequence-level watermarking method that leverages sequence entropy and subspace semantic differentiation to avoid region collapse and achieve highly imperceptible watermarking in low-entropy constrained generation tasks.

**Strengths:**

1. The paper effectively identifies and tackles the poor watermark performance under low-entropy constrained generation by analyzing sequence entropy in the continuous embedding space. It also introduces the novel region collapse phenomenon and proposes a principled design to mitigate it.
2. The use of LSH partitioning on a transformed high-quality subspace is conceptually elegant and ensures the watermark remains semantically hidden while maintaining output naturalness.
3. Written perspective is very clear and formulation is great.

**Weaknesses:**

1. Mean-centering does not fundamentally alter angular relationships in the embedding space; it merely shifts the cluster mean. Thus, the claim that LSH over the transformed subspace eliminates collapse is not theoretically substantiated. The increase in region entropy could be stochastic, not a structural guarantee.
2. The mean-centering transformation is purely heuristic. The paper provides no theoretical justification or citation for why subtracting the sample mean should effectively decorrelate embeddings or prevent region collapse.
3.  SeqMark’s detection procedure depends on re-sampling from the LM to re-estimate the mean embedding, introducing stochasticity and potential non-reproducibility. Even small differences in temperature, sampling seed, or model checkpoint may shift the embedding manifold, leading to inconsistent detection. The paper does not quantify this sensitivity or propose a robust verification protocol.

**Questions:**

Will the reproduction of results be stable so that maintaining an effective watermark checking progress?

---

> ### Author Response · Authors · 2025-11-21
> **Author Response to Reviewer YEPP**
>
> We thank you for the constructive review. We especially appreciate your’s positive comments on the strengths of our work, in particular the region collapse problem formulation for watermarking constrained generation outputs, as well as the effective solution SeqMark. We address your’s concerns and questions below:
>
> 1. We empirically demonstrated in Table 1 that the mean-centering transformation does alter the angular relationships between generations– it decreases the average pairwise cosine (which is equivalent to increase in angular distance). This transformation thus increases the region entropy, allowing for more effective semantic watermarking. To further support our claim, we provide the following theoretical justification that mean-centering decreases pairwise cosine similarity and increases detection accuracy: https://anonymous.4open.science/r/semantic_watermark-8BC4/SeqMark_Supplementary_Materials.pdf
>
> 2. As mentioned in question 1, we provide a theoretical justification that mean-centering leads to decrease in angular relationships and increase in watermark accuracy: https://anonymous.4open.science/r/semantic_watermark-8BC4/SeqMark_Supplementary_Materials.pdf. In addition, mean-centering transformation allowed us to experiment with a simple and effective solution. We acknowledge that there could be other principled methods to address region collapse, which we leave to future work to explore.
>
> 3. SeqMark detection does require exact sampling hyperparameters from generation in order to achieve the best performance. This arises from SeqMark requiring white-box access to the language model during detection, which although a bit cumbersome due to bookkeeping, is not entirely unreasonable. For example, prior work such as SWEET http://arxiv.org/abs/2305.15060, a popular tokenwise entropy-based watermaking algorithm, also requires whitebox access to the model and has similar detection-time requirements as our setup. Furthermore, we attempted to address this challenge with an alternative solution in our paper, Fast-SeqMark (Section 8.2). In the worst case, we believe it is not unrealistic to do hyperparameter sweep to match the  during detection. As mentioned above, we envision other whitebox-free methods for solving this problem in the future work.

---

### Note · Authors · 2026-01-06

I have read and agree with the venue's withdrawal policy on behalf of myself and my co-authors.